# Celebrities and Breast Cancer: A Multidimensional Quali-Quantitative Analysis of News Stories Shared on Social Media

**DOI:** 10.3390/ijerph19159676

**Published:** 2022-08-05

**Authors:** Priscila Biancovilli, Lilla Makszin, Faten Amer, Alexandra Csongor

**Affiliations:** 1Doctoral School of Health Sciences, University of Pécs, 7621 Pécs, Hungary; 2Institute of Bioanalysis, Medical School, University of Pécs, 7624 Pécs, Hungary; 3Szentágothai Research Center, University of Pécs, 7624 Pécs, Hungary; 4Department of Languages for Biomedical Purposes and Communication, Medical School, University of Pécs, 7624 Pécs, Hungary

**Keywords:** breast cancer, social media, content analysis, prevention, online, communication

## Abstract

In 2020, breast cancer was the most frequent type of cancer in 158 countries. To advise the population about risk factors and the importance of preventive measures, celebrities can be of great help, acting as spokespersons for reliable scientific information. The goal of this study is to analyse the content of news stories about breast cancer shared on different social media, examining how stories with celebrity presence are constructed. We performed a quali-quantitative multidimensional analysis of news stories in English that addressed breast cancer on the following social media platforms: Facebook, Pinterest, Twitter and Reddit. We compared news stories with and without the presence of celebrities. Our sample consists of 1594 news stories that had at least 1000 total shares across all analysed social media; 262 news stories mention a celebrity (16.44%), while 1332 stories do not (83.56%). Nevertheless, the ones that feature celebrities are, as a rule, more shared. The percentage of stories with celebrities addressing breast cancer prevention is quite low (1.9%). The same can be said for mentions of scientific papers/specialist quotes (3.4%). This research may help outline some possible paths that healthcare organizations and communication professionals can take to improve breast cancer content available online.

## 1. Introduction

The study of celebrities is relatively new, having started in the last decades of the 20th century [1,2], and developed in the 21st century. When we think about the term “celebrity”, we see that there is no consensual definition in the literature for this social phenomenon. Turner [3], whose book is a reference on the subject, explains that celebrities often emerge from entertainment or sports industries and attract huge media attention not only for their professional activities but also for their private lives. Another book [4] defines a celebrity as “not simply a noun but an adjective that signifies that someone possesses the quality of attracting attention”. Consequently, there are celebrities in different professions: journalists, lawyers, chefs, physical education professionals, and hairdressers, among others.

With the popularization of the Internet and social networks in the first decade of the 21st century, we witness the emergence of a new type of celebrity: social media influencers. Those individuals make use of tweets, blogs, videos or text in one or more social networks, inspiring audience attitudes across a myriad of areas [5]. Often, celebrities from the film industry, pop music or sports also make continuous use of social media to reach an even larger audience. On Facebook, the social network with the greatest number of monthly active users in the world as of February 2022 (2.91 billion) [6], 4 of the 10 fan pages with the most followers are of celebrities: Cristiano Ronaldo, football player (122.28 million fans); Shakira, pop singer (100.03 million fans); Vin Diesel, actor (96.71 million fans); and Leo Messi, football player (90.16 million fans) [7].

Regardless of their field of activity, celebrities have the power to influence a large number of people, helping to shape their opinions and behaviours [8,9]. Celebrities can also influence decisions that affect our health, both positively and negatively. Some studies indicate that celebrities can serve as agents of positive social change, prompting information-seeking about risk factors or early detection and fostering interpersonal communication [10,11,12].

In 2020, breast cancer was the most frequent type of cancer in 158 countries across the world; in absolute numbers, 2.2 million people were diagnosed with breast cancer, and 684 thousand died from the disease [13]. By the end of 2020, 7.8 million women had received a breast cancer diagnosis in the past 5 years [14].

Despite the expected growth in the number of diagnosed cases of breast cancer in the coming decades, changes in the population’s habits and lifestyle can reduce the probability of disease onset. Breast cancer often shows no symptoms in its early phases, and for this reason, many cases are detected at more advanced stages, when the chances of cure are lower [15]. Additionally, approximately 23% of breast cancer cases are preventable [16]. The adoption of some healthy lifestyle habits can help prevent breast cancer (and other types of cancer), such as exercising regularly, avoiding smoking, having a healthy and balanced diet and limiting the intake of alcoholic drinks [17].

There are case studies in the scientific literature about celebrities with cancer and their impact on the audience [11,18,19,20,21]. One of the most studied cases in recent years was that of the North American actress Angelina Jolie, who underwent a double mastectomy surgery to reduce her chances of developing breast cancer, as she has the faulty BRCA1 gene [18]. In the days and weeks after her announcement, the number of searches for tests that verify the presence of the BRCA gene and the number of preventive mastectomies multiplied in several countries (the so-called “Angelina Jolie effect”) [22,23].

Some health professionals noted that individuals seek medical examinations in greater numbers after a famous person’s medical announcement is revealed. A study that analysed celebrity breast cancer diagnoses and population screening behaviours over a 19-year period [24] concluded that there is a direct relationship between media coverage of celebrities with breast cancer and the increase in searches for mammograms or other breast examinations. According to this study, every 100 news stories published about a celebrity’s cancer increase the number of women getting mammograms by 0.77%.

Another investigation that measured the impact of celebrities on health-related knowledge and population behaviour observed 14 mechanisms explaining celebrity influence [25]. Among them are social networks, which allow their messages to reach a wide audience; commodification and social capital, wherein people follow celebrity medical advice to gain social status, and social constructivism, that is, when medical advice given by celebrities changes the way people interpret and perceive health information.

Despite existing discussions about the influence of celebrities on public health, to our knowledge, there are no studies that analyse celebrity content related to breast cancer shared on social media, comparing the characteristics of news stories with and without these personalities. Understanding how these narratives are currently constructed is the first step to plan and accomplish the desired results.

To decrease the number of breast cancer cases diagnosed at an advanced stage and to inform the population about risk factors and the importance of preventive measures against the disease, we believe that celebrities can be of great help, acting as spokespersons for reliable scientific information.

Therefore, the objective of this study is to analyse the content of news stories about breast cancer that are among the most shared on different social media, investigating from different angles how stories with celebrity presence are constructed. We will also compare the characteristics of these stories related to breast cancer with others that do not mention celebrities in the same time frame.

## 2. Materials and Methods

### 2.1. Data Collection

This is an exploratory analysis, without prior hypotheses. We examined news stories in English that addressed breast cancer. Data collection was conducted between 17 June 2019 and 17 June 2020.

For data collection, we used an online tool called BuzzSumo [26] to collect and organize the news stories used in our investigation. This tool scans an immense amount of social media data with over 5 billion articles, and it is updated in real time. We searched for “breast cancer” in quotation marks so that the results only showed us news stories that present the term written in this exact way, without considering the words written unconnectedly. The tool displays engagement metrics of articles, blog posts and videos shared on the following social media platforms: Facebook, Pinterest, Twitter and Reddit. Together, they add up to more than 3.6 billion monthly active users worldwide [27].

Our investigation was restricted to pages in English, with no country limitations. We comprehensively analysed the stories that had at least 1000 total shares. This selection criterion was set up for the following specific reasons: (1) The most shared news stories were those with higher visibility in the period under study, which makes this sample more relevant to our research. (2) We needed to establish a selection criterion for this sample that made content analysis feasible. (3) We believed that this selection criterion was sufficient to allow for a broad analysis of breast cancer narratives in the selected period.

BuzzSumo’s list of news stories was extracted to an Excel table with the following columns: total shares (sum of shares, i.e., when users share content on their personal pages, across all analysed social networks), total Facebook shares, total Twitter shares, total Pinterest shares, total Reddit engagements and published date. For this analysis, we only examined the total number of shares across the aforementioned social networks.

### 2.2. Content Analysis

To infer knowledge from a collection of qualitative data, we used the content analysis approach [28]. First, all of our corpus was coded. Codes for each news story were assigned according to a coding schedule (see Table 1). The second phase involved categorization. We regarded each news story as a unit of our corpus. In the coding schedule, the columns were categorized according to a number of different dimensions (see Table 1). Two researchers worked on analysing the material and sorting it. Both parties first contributed to the determination of the coding schedule and its dimensions. After that, each of them conducted their own independent analysis of a sample of one hundred news stories. The percentage of agreement between raters was utilized as the basis for the calculation of interrater reliability, and the resulting value was 83%. When this preliminary analysis was completed, one of the researchers categorized the rest of the corpus. The interpretation process was the final stage, at which researchers drew conclusions from the corpus.

### 2.3. Celebrity Presence

If the news story has as its focus an event involving a celebrity or only mentions a celebrity without this being the main theme of the text, we considered the celebrity presence to exist. We compared the stories with and without the presence of celebrities in relation to the other categories listed in Table 1.

### 2.4. Content Type

The classifications established by the analysts for “content type” were the following:Real-life story: when people with cancer and their family members share their stories or any other narrative from real life.Risk factors: when the text mentions a risk factor for breast cancer, such as smoking or sedentary lifestyle, among others.Treatment: stories that mention any type of treatment for breast cancer.New technology: stories that publicize or explain new technologies that can improve the detection or treatment of breast cancer.Solidarity: stories that mention solidarity actions, for instance, blood/hair donation for a sick person or when parties become involved in breast cancer awareness actions.Educational: news stories that elucidate what kind of food/behaviours can reduce the chances of developing cancer or what the symptoms of breast cancer are.Complaint: reports of difficulties that breast cancer patients suffer, for instance, problems with health insurance providers or the lack of medicines in clinics.Opinion: when writers express their opinions on any themes related to breast cancer, such as awareness campaigns.

### 2.5. Mentions of Breast Cancer Prevention, Early Detection or Screening Exams

We were keen to verify the number of news stories in our sample mentioning breast cancer screening and early detection, as it is of vital importance that the population gets informed about those topics. Many studies indicate that the knowledge of women influences their acceptance of treatment and screening exams [29,30,31]. It is also imperative to inform the lay audience that early-stage breast cancer does not always produce symptoms, and this is why mammograms should be incorporated into every woman’s health check-up during a certain period of their lives [32].

In addition to having knowledge about the screening examinations and the importance of early detection, prevention is also part of an effective strategy to reduce the incidence of this disease. Approximately 5% to 10% of all breast cancer cases are due to inherited mutations of the BRCA1 and BRCA2 genes; however, obesity, lack of physical activities and use of alcohol are also considerable risk factors that the population needs to be made aware of [33].

### 2.6. Mentions of Scientific Papers or Science Specialists

We also enumerated the number of times a news item in our sample mentioned a scientific article or quoted an expert. To reduce the spread of misinformation, having health professionals available to engage in dialogue with the media or create content on social networks is crucial [34].

### 2.7. Sentiment Analysis

Sentiment analysis and opinion mining were another part of the study, which investigated people’s opinions, feelings, evaluations, attitudes and emotions from written language [35]. It used natural language processing (NLP), machine learning and other data analysis techniques to examine text and provide quantitative metrics. We aimed to understand the prevailing sentiment towards breast cancer news on social media and whether this sentiment varied by content type. We used a tool called MonkeyLearn [36] to conduct the analysis, which provides free NLP-based sentiment analysis.

### 2.8. Statistical Analyses

Statistical analysis was performed using SPSS Statistics Version 28.0 (IBM, Chicago, IL, USA, 2021) and Microsoft Office Excel version 16 (Microsoft Corporation, Redmond, WA, USA, 2015). Differences in categorical variables were evaluated by Pearson’s chi-squared test. The Mann-Whitney U test was used for the comparison of the number of shares of stories to determine statistically significant parameters (*p* < 0.05).

## 3. Results

Screening the media for breast cancer news stories published between June 2019 and June 2020 resulted in 9811 hits. Of these, 1594 news stories had at least 1000 total shares and entered our analysis; 262 news stories mentioned a celebrity (16.44%), while 1332 stories did not (83.56%).

When we observe the mean rank of shares (across all social media sites) of news stories with and without celebrities (Table 2), we see that news citing a celebrity is shared more frequently.

Moreover, there is a very strong statistical connection between content type and celebrity presence (Table 3). The themes in which celebrities usually appear are different from those in which these people are not present.

In our sample, there was no presence of celebrities in the themes of “complaint”, “new technology” and “opinion”. Celebrities only appeared once each in the themes “risk factors” and “treatment”. That is, stories about risk factors without celebrities were 30.3 times more frequent than those with celebrities, and stories about treatment without celebrity mentions appeared 28.8 times more frequently in our sample than those with celebrities.

We also observed statistical significance when analysing mentions of scientific papers or quotes from health experts in news stories with and without celebrities (Table 4). In stories without celebrities, 36.8% of the cases mention scientific papers or quotes from experts. However, if celebrities were involved, only 3.4% of the stories mentioned scientific articles or quoted specialists (10.8× less frequent).

Regarding the prevailing sentiment in our sample (Table 5), we observed that news stories with celebrity presence tended to be more negative (2.7× more) than those without; also, there were 2.7 more stories classified as neutral without celebrities. Finally, positive stories without mentions to celebrities were 1.3 times more prevalent than those with famous personalities.

The celebrity story with the highest number of shares in our sample (*n* = 356,393) had the following title: “Shannen Doherty reveals breast cancer is back, now stage 4”. Doherty is a North American actress, and this is an example of a text with a prevalence of negative sentiment. There are no mentions of early diagnosis or prevention, nor are there any links to scientific articles or quotes from health experts. The same story is featured on another website, also with a high number of shares (*n* = 138,195). Among the 20 most shared stories, we found another one involving a celebrity: “Sad News, Robin Roberts [a North American television broadcaster] Painfully Reveals She Had Breast Cancer” (*n* = 134,769). Again, the prevailing sentiment is negative, and there is no mention of screening exams or prevention. In contrast, the most shared story in our sample (*n* = 1,822,993) had a positive sentiment and did not mention celebrities: “Trial vaccine wipes out breast cancer in Florida patient”.

We also identified a statistically significant difference (chi-square test: *p* = 0.011; Cramer’s value = 0.064) regarding mentions of breast cancer prevention in news stories with and without celebrities. There were three times as many stories without celebrities addressing prevention (5.7%) as there were with celebrities (1.9%).

There was a moderate statistical connection (chi-square test: *p* < 0.001; Cramer’s value = 0.105) when we compared news stories that addressed early detection or screening exams with and without the presence of celebrities. There were twice as many stories without celebrities addressing early detection/screening exams (21.5%) as there were with celebrities (10.3%).

## 4. Discussion

This analysis highlighted that, although the majority of the stories in our sample did not mention celebrities (83.56%), the ones that featured celebrities were, as a rule, more shared. Nevertheless, it is notable that, in our sample, the percentage of stories addressing breast cancer prevention was quite low, and it was even smaller when there was any mention of a celebrity. The same can be said for early detection, screening exams and mentions of scientific papers/specialist quotes. Although the proportion of stories containing these subjects was higher, when combined with the presence of celebrities, the frequency dropped considerably. A practical implication of the mentioned findings is that there should be more breast cancer news stories linking celebrity speech to scientifically sound messages on prevention, screening exams and early diagnosis. This increases the possibility that the content will be more resonant with target audiences.

In relation to cancer communication carried out by celebrities, there are reports in the literature about the impact it has on the lay public. In addition to the previously mentioned “Angelina Jolie effect”, it is also worth mentioning that when the Australian singer Kylie Minogue was diagnosed with breast cancer, improvements in women’s cancer prevention behaviours were observed in Australia. When compared to the previous six months following the artist’s diagnosis, the number of women aged 25–44 years who had screening exams increased by 20% [37]. In addition, there was a 20-fold increase in media coverage of breast cancer in Australia, highlighting that early detection is important for a good prognosis and that young women can also be affected by the disease [38].

A cross-sectional survey found that approximately 40% of Black women talked about cancer following the death of singer Aretha Franklin, and more than 50% intended to talk about the disease in their social circles [11].

A similar logic applies to other contexts in the health area. During the COVID-19 pandemic, the largest number of likes was received by tweets of celebrities and politicians, outperforming those coming from health and scientific institutions [39]. This attention, however, can have deleterious effects on the health of the population. For instance, in 2021, pop star Nicki Minaj declared to her 22.8 million followers on Twitter that she was against the administration of the COVID-19 vaccine, claiming it causes sexual impotence; her message was shared more than 26,000 times and likely led some people not to get vaccinated [40].

Regarding our sample, it is also worth noting that most of the stories that mention celebrities had a negative sentiment (53%). However, the number was much lower for stories that did not mention celebrities (19.7%). Messages and posts that elicit positive emotions from the audience are more favoured on social media [41,42,43]. Research indicates that posts that provoke positive feelings in the audience, such as laughter and amusement, have a greater likelihood of being shared more [44]. Thus, another practical implication could be achieved if news stories featured celebrities addressing optimistic topics, such as the latest advances in science and technology, real stories of hope and motivation or science-based cancer prevention strategies, to name just a few examples of topics that are rare in our sample.

Social media is a potential venue for interventions to reach a diverse audience, which may not be achievable through traditional approaches [45], but it is of fundamental importance that these messages are better designed to reach the target audience in an effective way. Media coverage of famous people can motivate knowledge-seeking activity [20] and may also positively impact the use of health services [46], according to the type of information that is shared. For this, online health content should be developed in a more strategic and persuasive way, using celebrities more frequently as a bridge to efficiently inform and educate the audience about symptoms, early detection and prevention strategies of breast cancer.

Our study has some limitations. Firstly, in the content analysis, we should mention the limited number of investigated news stories, as we did not have enough resources to qualitatively analyse thousands of articles whilst maintaining the quality of the process. Therefore, there is a possibility that the results obtained in the analysis of our sample are different from those that we would obtain if the entire corpus were included in the study. Secondly, there is the fact that our sample is limited to news stories in English. If we investigated other languages, we might have encountered variations in the way celebrities are portrayed in breast cancer stories. Consequently, we consider that it is not possible to generalize the results observed in this article to all languages and cultural settings.

There are some avenues for future research in this area. The main one involves the analysis of a longer period, so that we can see whether the same trends hold or whether there are differences. We also believe that it is interesting to analyse online content about breast cancer in other languages, so that the results and consequent communication interventions in social media will be tailored according to each environment.

## 5. Conclusions

Breast cancer is one of the most prevalent types of cancer worldwide. Many of the cases can be prevented or, at least, diagnosed early, increasing the chances of cure. For this, the population needs to be better informed about symptoms, screening exams and treatment possibilities, among other aspects. Using social media effectively to transmit scientifically accurate information about this disease is a compelling strategy. In this sense, celebrities can be of great help, channelling the public’s attention and increasing the reach of these messages. The results of this research may help outline some possible paths that healthcare organizations and communication professionals can take to improve breast cancer content available online.

## Figures and Tables

**Table 1 ijerph-19-09676-t001:** Coding manual, including the coding schedule and its categories.

Celebrity Presence	Content Type	Mentions of Breast Cancer Prevention	Mentions of Early Detection or Screening Exams	Sentiment	Mentions of Scientific Paper or Specialist Quotes
Yes	Real-life story	Yes	Yes	Positive	Yes
No	Risk factors	No	No	Negative	No
	Treatment			Neutral	
	New technology				
	Solidarity				
	Educational				
	Complaint				
	Opinion				

The column headings indicate the dimensions to be coded.

**Table 2 ijerph-19-09676-t002:** Total and mean number of shares of stories that feature celebrities versus stories without celebrities.

Celebrity	*N*	Mean of Shares	Sum of Shares
No	1332	784.63	104,513,050
Yes	262	862.92	22,608,450
Total	1594		

Mann–Whitney U test: *p* = 0.012.

**Table 3 ijerph-19-09676-t003:** Celebrity presence in our sample according to content type.

	Celebrity Presence		
Content Type	No	Yes	Total
Complaint	14	0	14
Educational	94	3	97
New technology	82	0	82
Opinion	6	0	6
Real-life story	627	209	836
Risk factors	161	1	162
Solidarity	195	48	243
Treatment	153	1	154
Total	1332	262	1594

Chi-square test: *p* < 0.001; Cramer’s value = 0.293 (very strong connection).

**Table 4 ijerph-19-09676-t004:** Frequency that links to scientific papers and quotes from health specialists appearing in news stories.

			Celebrity		Total
			**No**	**Yes**	
Link to Scientific Paper or Specialist Quote	No	Count	842	253	1095
		Expected Count	9150	1800	10,950
		%	63.2%	96.6%	68.7%
	Yes	Count	490	9	499
		Expected Count	4170	820	4990
		%	36.8%	3.4%	31.3%
Total		Count	1332	262	1594
		Expected Count	13,320	2620	15,940
		%	100.0%	100.0%	100.0%

Chi-square test: *p* < 0.001; Cramer’s value = 0.267 (very strong connection).

**Table 5 ijerph-19-09676-t005:** Frequency of each sentiment in news stories with or without celebrities in our sample.

			Celebrity		Total
			No	Yes	
Sentiment	Negative	Count	263	141	404
		Expected Count	3376	664	4040
		%	19.7%	53.8%	25.3%
	Neutral	Count	547	40	587
		Expected Count	4905	965	5870
		%	41.1%	15.3%	36.8%
	Positive	Count	522	81	603
		Expected Count	5039	991	6030
		%	39.2%	30.9%	37.8%
Total		Count	1332	262	1594
		Expected Count	13,320	2620	15,940
		%	100.0%	100.0%	100.0%

Chi-square test: *p* < 0.001; Cramer’s value = 0.300 (very strong connection).

## Data Availability

The datasets used and/or analysed during the current study are available from the corresponding author on reasonable request.

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
