# Peer review of "Celebrities and Breast Cancer: A Multidimensional Quali-Quantitative Analysis of News Stories Shared on Social Media"

_ijerph, 2022, doi:10.3390/ijerph19159676_

Round 1
Reviewer 1 Report
Spontaneous revelation of a disease or disability by celebrities can be understood by ordinary people as testimony, invitation to prevention, positive narration, further sharing of a life already participated and followed by the community. Those who talk about themselves feel less alone, they give voice to their pain and a thousand fears. Sharing own experiences can drive hope and make everything more acceptable. According to experts, the benefits are different. Posts on social media become a way to keep in touch with friends and have new ones, to get naked in a moment of fragility, avoid isolation and receive messages of strength and hope. But they also become a means of circulating correct information on diagnoses, treatments and therapies. All this using simple and immediate language.
The paper has a potential interest. A revision of the discussion is suggested to understand how the results of this study may be useful in outlining possible paths that healthcare organizations and communication professionals can take to improve breast cancer content available online.
Author Response
Please see the attachment.
Response to Reviewer 1 Comments
Spontaneous revelation of a disease or disability by celebrities can be understood by ordinary people as testimony, invitation to prevention, positive narration, further sharing of a life already participated and followed by the community. Those who talk about themselves feel less alone, they give voice to their pain and a thousand fears. Sharing own experiences can drive hope and make everything more acceptable. According to experts, the benefits are different. Posts on social media become a way to keep in touch with friends and have new ones, to get naked in a moment of fragility, avoid isolation and receive messages of strength and hope. But they also become a means of circulating correct information on diagnoses, treatments and therapies. All this using simple and immediate language.
The paper has a potential interest. A revision of the discussion is suggested to understand how the results of this study may be useful in outlining possible paths that healthcare organizations and communication professionals can take to improve breast cancer content available online.
Response 1:
We truly appreciate the suggestions and are grateful for the positive feedback from Reviewer 1. We have better structured the discussion, so that readers can better understand how the results of this study may be useful in outlining possible paths that healthcare organizations and communication professionals can take to improve breast cancer content available online. Please see lines 294-297, 329-334, 340-343. This is the line count in "All markup" review mode in Word.

Reviewer 2 Report
I thank you for the opportunity to review the article entitled: “Celebrities and Breast Cancer: A Multidimensional Quali-Quantitative Analysis of News Stories Shared on Social Media”. The article is interesting and deals with a current issue, that is, the content of stories about breast cancer shared on different social media, examining how stories with celebrity presence is constructed.
Although the article is beneficial for the literature, below I propose some suggestions that I hope will help the authors to improve their work.
Introduction. The introduction is clear and the reader feels comfortable entering into the issues discussed. The structure of the introduction starts with a general definition, then highlights the gaps in the literature (with recent supporting citations), and finally proposes the research questions (see Grant, A. M., & Pollock, T. G. (2011). Publishing in AMJ—Part 3: Setting the hook). I like this approach and I like the introduction as it is set up. I note one problem, however: the assumption on which the paper is based is that there are no studies on the same topic. Yet there is a very similar paper (i.e., Döbrössy, B., Girasek, E., Susánszky, A., Koncz, Z., GyÅ‘rffy, Z., & Bognár, V. K. (2020). " Clicks, likes, shares and comments" a systematic review of breast cancer screening discourse in social media. PloS one, 15(4), e0231422). How does your study differ from previous literature? We need to act on this to substantiate the need for a study like this. The absence of a theory paragraph (see commentary below) does not help the reader to understand whether or not this study has utility compared to the literature on the subject.
Theory and Hypotheses. The authors state that since this is an exploratory study, there are no hypotheses. While I agree with this approach, I feel very uneasy as a reader if I cannot find a paragraph of theory, from which some propositions-even indirect ones-can emerge. I urge the authors to write a paragraph of theoretical background from which the need to carry out a study like this can arise. Moreover, I just wonder one thing: in a study that assumes that many scholars are talking about this phenomenon, is it ever possible that an article submitted in July 2022 uses data from more than 2 years earlier? There is a need for authors to update the literature on the topic, even inserting themselves into a recent discourse initiated by other scholars.
Method. The methodology is really good. The authors clearly explain the steps they followed. I just have my doubts that a dataset from 2019-2020 cannot best represent the reality in which we live today, as the world (and even the world of social media) has seen considerable upheaval. In fact, as I reread the study, I realize that this is the main (strong) perplexity I have about the study. The problem may be only in the fact that we are talking about a phenomenon that may have changed completely as a result of covid (we are seeing how young people, but also older generations are changing attitudes toward social media), so I don't know whether this study can be considered current or already obsolete. Therefore, I would invite the authors to update their content analysis to include data for the last 2 years (or at least for the last year).
Results. I applaud the authors for the results, which seem clear and rigorous to me.
Discussion, Limitations and Future Research. The discussion is really good. I would prefer more structuring of the Discussion, so that both the theoretical and practical implications of the study can be clearly identified – the latter, in particular, are virtually absent. The limitations are okay, and also are the implications for future research.
I invite the authors to work on the manuscript further to improve its contribution to the literature. I hope that my suggestions could serve this purpose.
Author Response
Please see the attachment.
Response to Reviewer 2 Comments
Point 1: Introduction. The introduction is clear and the reader feels comfortable entering into the issues discussed. The structure of the introduction starts with a general definition, then highlights the gaps in the literature (with recent supporting citations), and finally proposes the research questions (see Grant, A. M., & Pollock, T. G. (2011). Publishing in AMJ—Part 3: Setting the hook). I like this approach and I like the introduction as it is set up. I note one problem, however: the assumption on which the paper is based is that there are no studies on the same topic. Yet there is a very similar paper (i.e., Döbrössy, B., Girasek, E., Susánszky, A., Koncz, Z., GyÅ‘rffy, Z., & Bognár, V. K. (2020). " Clicks, likes, shares and comments" a systematic review of breast cancer screening discourse in social media. PloS one, 15(4), e0231422). How does your study differ from previous literature? We need to act on this to substantiate the need for a study like this. The absence of a theory paragraph (see commentary below) does not help the reader to understand whether or not this study has utility compared to the literature on the subject.
Response 1:
We really appreciate the literature shared with us and added it as a reference in our introduction (see lines 67-71). This is the line count in "All markup" review mode in Word. The aforementioned study, however, is different from ours, as well as other articles cited in this investigation and found in the literature. To our knowledge, there are no studies so far that analyse specifically celebrity content related to breast cancer shared on social media, comparing the characteristics of news stories with and without these personalities.
Point 2: Theory and Hypotheses. The authors state that since this is an exploratory study, there are no hypotheses. While I agree with this approach, I feel very uneasy as a reader if I cannot find a paragraph of theory, from which some propositions-even indirect ones-can emerge. I urge the authors to write a paragraph of theoretical background from which the need to carry out a study like this can arise. Moreover, I just wonder one thing: in a study that assumes that many scholars are talking about this phenomenon, is it ever possible that an article submitted in July 2022 uses data from more than 2 years earlier? There is a need for authors to update the literature on the topic, even inserting themselves into a recent discourse initiated by other scholars.
Response 2: We agree that the addition of a theory paragraph makes the article more interesting and well-grounded. This is why we have added it (please see lines 68-91). We have also added more updated literature on the topic, as suggested. For this, please check lines 54, 68, 77, 318, 323, 327.
Point 3: Method. The methodology is really good. The authors clearly explain the steps they followed. I just have my doubts that a dataset from 2019-2020 cannot best represent the reality in which we live today, as the world (and even the world of social media) has seen considerable upheaval. In fact, as I reread the study, I realize that this is the main (strong) perplexity I have about the study. The problem may be only in the fact that we are talking about a phenomenon that may have changed completely as a result of covid (we are seeing how young people, but also older generations are changing attitudes toward social media), so I don't know whether this study can be considered current or already obsolete. Therefore, I would invite the authors to update their content analysis to include data for the last 2 years (or at least for the last year).
Response 3: Although we understand the reviewer's point of view, we really believe that an update of the study is not necessary at this time, for a few reasons. This type of analysis is time demanding. We needed approximately 1 year to complete the collection, categorization and analysis of news stories from our corpus. For the scientifically rigorous update with more recent data, we would need at least another year to complete the analysis. We agree that the world is dynamic and so are social networks, which is why we also include, in the discussion, articles related to social networks usage and celebrities during the COVID-19 pandemic. For this, please see lines 316-323. It is also worth mentioning that our corpus covered 4 months of the COVID-19 pandemic, from March (when WHO declared COVID-19 a pandemic) to June 2020. Moreover, research on the pandemic so far has underlined our findings and confirmed that health professionals need to be strategic and proactive in engaging with health consumers on social media if they hope to counteract the harmful effects of misinformation.
Point 4: Discussion, Limitations and Future Research. The discussion is really good. I would prefer more structuring of the Discussion, so that both the theoretical and practical implications of the study can be clearly identified – the latter, in particular, are virtually absent. The limitations are okay, and also are the implications for future research.
Response 4:
We agree that a better structuring of the discussion would be benefitial to our article. This is why we have made some modifications to the text. Please see lines 294-297, 329-334, 340-343.
I invite the authors to work on the manuscript further to improve its contribution to the literature. I hope that my suggestions could serve this purpose.
Response 5:
All authors are profoundly grateful for Reviewer’s 2 careful reading and constructive comments to our work.

Round 2
Reviewer 2 Report
I thank you for the opportunity to read the new version of this paper, as I believe the authors have worked hard to improve what was already a very interesting paper.
First, with reference to the introduction, it now seems to be clearer what difference this study proposes from the literature on the topic.
With reference to the theory, although I appreciate more if there is an ad hoc theoretical framework paragraph, what has been included in the introduction is also useful and especially functional for a reader to understand the reference theory.
I realize that it was not easy to update the methodology and analysis with the last two years, but I am satisfied with the authors' response and references to this point in the limitations of the study.
In conclusion, I am satisfied with the changes made and consider this study worthy of publication. Good luck to the authors.